# Promoting Beneficial Arthropods in Urban Agroecosystems: Focus on Flowers, Maybe Not Native Plants

**DOI:** 10.3390/insects14070576

**Published:** 2023-06-23

**Authors:** Stacy M. Philpott, Azucena Lucatero, Sofie Andrade, Cameron Hernandez, Peter Bichier

**Affiliations:** 1Environmental Studies Department, University of California, Santa Cruz, CA 95064, USA; 2Ecology and Evolutionary Biology Department, University of California, Santa Cruz, CA 95064, USA

**Keywords:** garden, biodiversity, invasive species, native species, bees, spiders, ants, ladybeetles, urbanization

## Abstract

**Simple Summary:**

Urban sprawl contributes to biodiversity loss, but the presence of native plants in urban areas may help to support diversity. In urban gardens, where non-native plants are common, the role of native plants may be especially important in providing resources to pollinators and other beneficial arthropods, like natural enemies of pests, but little research has examined how native plants affect non-pollinators in gardens. We sampled plants, bees, and three groups of natural enemies (ladybeetles, ants, and ground-foraging spiders) in gardens in California. We found that native plants represented about 10% of the species and only 2.5% of the plants found. The native plants present did not have large impacts on the numbers or diversity of bees, ladybeetles, or ants, but did have an unexpected negative effect on non-native spiders. Other garden features, such as garden size, flowers, mulch, and trees and shrubs, did have mostly positive impacts on the biodiversity of bees, ants, and spiders, but the impacts of each feature differed by organism type. Natural habitat near to gardens was also important for native ants, non-native bees, and ladybeetles. In sum, native plants, when rare within gardens, likely have little benefit, but other garden features can be manipulated in order to promote beneficial arthropods in gardens.

**Abstract:**

(1) Urbanization threatens biodiversity, yet urban native plants support native biodiversity, contributing to conservation and ecosystem services. Within urban agroecosystems, where non-native plants are abundant, native plants may boost the abundance and richness of beneficial arthropods. Nevertheless, current information focuses on pollinators, with little attention being paid to other beneficials, like natural enemies. (2) We examined how the species richness of native plants, garden management, and landscape composition influence the abundance and species richness of all, native, and non-native bees, ladybeetles, ants, and ground-foraging spiders in urban agroecosystems (i.e., urban community gardens) in California. (3) We found that native plants (~10% of species, but only ~2.5% of plant cover) had little influence on arthropods, with negative effects only on non-native spider richness, likely due to the low plant cover provided by native plants. Garden size boosted native and non-native bee abundance and richness and non-native spider richness; floral abundance boosted non-native spider abundance and native and non-native spider richness; and mulch cover and tree and shrub abundance boosted non-native spider richness. Natural habitat cover promoted non-native bee and native ant abundance, but fewer native ladybeetle species were observed. (4) While native plant richness may not strongly influence the abundance and richness of beneficial arthropods, other garden management features could be manipulated to promote the conservation of native organisms or ecosystem services provided by native and non-native organisms within urban agroecosystems.

## 1. Introduction

Urbanization poses a major risk for plant and animal biodiversity, and may lead to increases in the abundance and richness of non-native plants, with important impacts for other organisms that inhabit urban green spaces. Urban sprawl is steadily increasing as urbanization intensifies worldwide, which has led to landscape modification, fragmentation, increased pollution, climatic changes, the degradation of ecological communities, and biodiversity loss [1,2,3,4]. Urbanization is also associated with the introduction of non-native species [5,6], a higher proportion of non-native species [7], and can lead to biotic homogenization, further threatening biodiversity [8,9,10]. Moreover, threats to biodiversity from the expansion of urban land are expected to intensify over the next 20 years, especially in tropical developing nations [11].

One important green space within urban systems is the urban agroecosystem. Urban agroecosystems can include community, allotment, and residential gardens, urban farms, and other locations within cities where vegetables, medicinal plants, herbs, and fruit trees are grown [12,13,14]. Urban agroecosystems substantially contribute to the global food supply [15,16]. Beyond food security contributions, urban agroecosystems can provide cultural connections for city residents [17,18,19] and support physical and mental health, connection to nature, and other aspects of human wellbeing [20,21,22,23]. Gardens also create critical green spaces in cities that provide habitat for biodiversity [12,24]. Many studies have explored the impacts of agroecosystem management (e.g., woody plant abundance, floral abundance and richness, ground cover characteristics) and landscape context on biodiversity and ecosystem within urban agroecosystems (e.g., [25]), while fewer studies focus on the impacts of native plants on native organisms in urban agroecosystems.

Although non-native species generally represent between 10 and 35% of all species in urban ecosystems [26], in urban agroecosystems, where many non-native plants are intentionally planted, the percentage of non-native organisms averages 55% [27], and for plants, this can be as high as 95% [18]. Such dramatic shifts in plant species composition and origin may affect the biodiversity of beneficial arthropods, like pollinators and natural enemies of pests, or the ecosystem services they provide within urban agroecosystems, and may also lead to negative impacts on ecosystem services if non-native species disrupt ecological interactions [6]. Thus, the role of native versus non-native plants for supporting biodiversity in urban agroecosystems should be carefully considered. 

Both native and non-native plants may play important roles in supporting native biodiversity in urban systems, but their impacts on the diversity of arthropods deserve further exploration and analysis. Native plants may better support arthropods (compared to non-native plants) because they are locally adapted to regional environmental conditions and because native perennials may provide year-round provision of resources to arthropods [28]. Notably, native plants in urban ecosystems may benefit the abundance, richness, and activity of pollinators [29], bee abundance [30], parasitoid longevity [31], and local animal biodiversity more generally [32]. Yet, there may be nuanced relationships between native plants and arthropods depending on arthropod taxon, the response variable examined, foraging preferences of arthropods, time since native plant establishment, and plant traits. Native plants may boost both bee abundance and richness [33], or bee abundance but not species richness [30], or may affect the species composition and abundance of non-native bee visits, but not overall bee visits [34]. Moreover, in a review of 16 studies comparing the attractiveness of native and non-native plants to bees, 44% found that bees were more attracted to native plants, 38% found that bees were more attracted to non-native plants, and 18% of studies found no difference [35]. Native plants boost invertebrate abundance more generally, but at different rates depending on invertebrate functional group; predators and herbivores were 23% and 17% more abundant, respectively, on native plants compared to introduced plants [36]. A recent meta-analysis documented that native plants supported higher animal diversity in 43% of the 70 included studies, but frequently had mixed (33% of studies) and neutral (17% of studies) effects [32]. Non-native plants in agroecosystems may initially attract predatory insects, with natives becoming more attractive as they become more established over time [37]. Finally, different native plant traits can be more or less attractive to different insect functional groups [38]. 

Most studies examining the relationship between native plants and animal biodiversity have focused on pollinators, with few looking at other groups, despite the fact that other arthropods utilize floral resources. Natural enemies of agricultural pests, like ladybeetles (Coleoptera: Coccinellidae) [39,40], ants (Hymentopera: Formicidae) [41], and spiders (Aranae) [42], may also utilize and benefit from floral resources. However, the impact of native plants on these beneficial arthropods has not often been examined. For instance, in a recent meta-analysis, pollinators made up 94% of invertebrates studied [32]. There has also been a limited focus on native arthropods, even though native arthropods can comprise a larger fraction of communities in gardens [35]. We aim to fill this research gap by taking a multi-taxa approach to examine how the presence and abundance of native plants may support one group of pollinators (bees) and three groups of natural enemies (ladybeetles, ants, and ground-foraging spiders) within urban agroecosystems. We also consider the native status of arthropods to see if all, native, or non-native arthropods respond to native plants in different ways. We focused our study on urban agroecosystems, a habitat frequently highlighted as important in supporting biodiversity in urban areas [12]. We specifically ask: (1) Does native plant richness support a higher abundance and species richness of all bees, ladybeetles, ants, and ground-foraging spiders in community gardens? (2) Does native plant richness support a higher abundance and species richness of native bees, ladybeetles, ants, and ground-foraging spiders in community gardens? and (3) Does native plant richness support a higher abundance and species richness of non-native bees, ladybeetles, ants, and ground-foraging spiders in community gardens? We hypothesized that native plants would have strong positive effects on the abundance and species richness of bees and ladybeetles, while having neutral or no effects on ant and ground-foraging spider abundance and species richness. Moreover, we expected that the richness of native plants would have stronger impacts on native organisms compared with all or non-native organisms. 

## 2. Materials and Methods

### 2.1. Study System

We conducted our research in 19 gardens in the California Central Coast region in Monterey, Santa Clara, and Santa Cruz counties (Figure 1). All gardens are community gardens where vegetables, fruit trees, and ornamental plants are grown. The gardens are managed either collectively or in individual plots or allotments, range in size from 444 m^2^ to 15,400 m^2^, and had been cultivated for between 4 and 46 years at the earliest year of the study. The gardens are all managed using organic practices, and none allow chemical pesticides or herbicides to be used. None of the gardens contain large patches of grass, and none experience regular mowing. Each garden was separated from other garden sites by at least 2 km and up to 90 km. 

### 2.2. Vegetation Sampling

In each garden, we sampled vegetation and ground cover as metrics of the local habitat management. At the center of each garden, we established a 20 m × 20 m plot within which we measured canopy cover with a concave vertical densiometer (at the center of the plot, and at each edge), counted and identified all trees and shrubs, and counted the number of trees and shrubs in flower. Within four randomly located 1 m × 1 m plots within the 20 m × 20 m plot, we sampled ground cover by visually estimating the percent cover of (a) bare ground, (b) grass, (c) herbaceous plants, (d) rocks/wood panels, (e) leaf litter, (f) mulch, and (g) concrete. In the same plots, we identified and estimated the percent cover of each herbaceous plant species, counted the number of flowers, and measured the height of the tallest herbaceous vegetation. We estimated the percent cover of grass, but did not identify grasses at the species level. Each of these vegetation and ground cover features were sampled six times in 2013 (between 17 May and 2 June, 18 and 24 June, 16 and 22 July, 12 and 21 August, 10 and 16 September, and 11 October and 19 November), six times in 2014 (between 17 and 20 June, 7 and 10 July, 27 and 30 July, 19 and 21 August, 8 and 10 September, and 29 September and 1 October), and six times in 2015 (between 13 and 15 May, 16 and 30 June, and 7 and 11 July, on 2 August, between 1 and 15 September, and between 21 September and 6 October) near to the dates that we sampled arthropods. Our dataset contained several plant morphospecies (<4% of total plant cover) that we were not able to identify at the species level, but these were not included in the analysis. 

### 2.3. Native Plant Characterization and Abundance

We classified all herbaceous plant species identified in vegetation surveys according to whether they are native to California. Specifically, we categorized herbaceous plant species as native or not native based on information from four plant databases (e.g., Calflora (https://www.calflora.org/ (accessed on 10 November 2022)), USDA Plants (https://plants.sc.egov.usda.gov/java/ (accessed on 10 November 2022)), Jepson eFlora (https://ucjeps.berkeley.edu/eflora/ (accessed on 10 November 2022)), and the Missouri Botanical Garden (https://www.missouribotanicalgarden.org/plantfinder/plantfindersearch.aspx (accessed on 10 November 2022)). For species of herbaceous plants for which we found conflicting information, we first followed information from Calflora, and then from other databases. For any herbaceous plants for which we could not find information, we assumed them to be non-native to California. To determine the total number of native herbaceous plant species in each site for each year, we used the cumulative number of native herbaceous species encountered across all sample plots and all time periods for that site. To determine the total percent cover of native herbaceous plants, we summed all herbaceous plant cover (minus grass) from all sample points in a site in a year, and calculated the fraction of that cover comprising native herbaceous plant species. 

### 2.4. Landscape Composition and Diversity

We used information from the 2011 National Land Cover Database (NLCD, 30 m resolution) [43] to measure land cover composition within 2 km of each garden. We extracted data for all NLCD land cover classes, and combined classes to create four landscape variables for this study: (1) natural habitat (which combined deciduous, evergreen, and mixed forests, dwarf scrub, shrub/scrub, and grassland/herbaceous cover classes); (2) open habitat (which combined lawn grass, parks, and golf courses); (3) urban habitat (which combined low-, medium-, and high-intensity developed land); and (4) agricultural habitat (which combined pasture/hay and cultivated crops). We excluded land cover types that did not comprise more than 5% of the surrounding land cover for any garden (e.g., open water, wetlands; and barren land). 

### 2.5. Arthropod Sampling

We sampled four groups of arthropods, namely bees (Apoidea), ladybeetles (Coleoptera: Coccinellidae), ants (Hymentopera: Formicidae), and spiders (Arachnida), and chose sampling methods recommended specifically for each of these groups. These arthropods are common in urban areas, vary in life history, and support ecosystem services within gardens [44]. We sampled arthropods between May and October in 2013, 2014, and 2015, and visited gardens five or six times each summer. We surveyed bees in 2013 and 2015, ladybeetles in 2014 and 2015, and ants and ground-foraging spiders in 2013. 

We sampled bees with elevated pan traps and aerial hand netting designed to specifically collect bees [45]. We constructed pan traps using 400 mL yellow, white, and blue plastic bowls painted with Clear Neon Brand and Clear UV spray paint. We mounted traps on 1.2 m PVC pipes and filled bowls with a water (300 mL) and soap (4 mL) solution [46]. We placed traps (one yellow, one white, one blue) 5 m apart from each other at the center of the 20 m × 20 m plot in each garden. We placed traps between 8 and 9 AM and collected them between 5 and 7 PM on the same day. We emptied traps into containers and transported them to the lab for sorting and pinning. We put out elevated pan traps six times in 2013 (29–31 May, 25–27 June, 23–25 July, 12–15 August, 17–20 September, 9–11 October) and four times in 2015 (6–13 April, 16–19 June, 8–10 July, and 11–14 August). We netted bees from flowers found within the 20 m × 20 m plot and within 20 m of the plot boundaries (for a total of a 60 m × 60 m plot) for 30 min on warm, sunny days. We collected bees six times during 2013 (17–22 May, 18–24 June, 16–22 July, 12–21 August, 10–11, 23 September, 11–15 October) and six times in 2015 (16–19 June, 7–10 July, 31 July–1 August, 11–14 August, 1–3 September, 21–24 September). We identified bees using online resources, image databases, books, and dichotomous keys [47,48,49,50], and by comparing specimens to bees in the Kenneth S. Norris Center for Natural History on the University of California, Santa Cruz campus. We identified all specimens to the highest taxonomic level possible, with more difficult groups identified to the morphospecies level. 

We sampled ladybeetles in the 20 m × 20 m plots at the center of each garden six times in 2014 (17–20 June, 7–10 July, 27–30 July, 19–21 August, 8–10 September, and 29 September–1 October) and five times in 2015 (16–19 June, 7–10 July, 31 July–1 August, 11–14 August, 1–3 September, and 21–24 September). We used visual surveys and sticky traps [51]. We visually surveyed and collected ladybeetles in eight randomly selected 0.5 m × 0.5 m plots within the 20 m × 20 m plots. Second, we placed four 7.62 cm × 12.7 cm yellow sticky strip traps (BioQuip Products Inc., Rancho Dominguez, CA, USA) on wire stakes placed in the ground next to vegetation at four random locations and collected them after 24 h. All lady beetles were identified at species level, or genus level when species identification was impossible (e.g., *Scymnus* sp. on sticky traps), using online resources (e.g., [52]) and identification guides [53]. 

We sampled ground-foraging spider (hereafter spider) and ant activity density with pitfall traps placed for 72 h in each site four times in 2013 (20–23 May, 17–20 June, 15–18 July, 11–14 August, and 9–12 September). We constructed pitfall traps from 350 mL plastic tubs (11.4 cm diameter × 7.6 cm deep). We placed traps in the middle of the 20 m × 20 m plot in each garden arranged in two rows of three traps, with every trap separated from others by 5 m. We buried traps at the level of the soil surface, and filled traps with 200 mL of a saturated saline solution with a drop of unscented detergent to break the surface tension. We placed green plastic plates (7.62 cm diameter) 7–8 cm over each trap to prevent the capture of non-target taxa (e.g., flies) and to limit overflow for traps from overhead irrigation (Appendix A). Upon collection, we rinsed arthropods with water, separated them by order, and then stored insects in vials with 70% ethanol. Adult spiders in common families were identified at species level, and other individuals of other families were identified at the morphospecies level using several guides [54,55,56,57,58]. Ants were identified at species level using an online guide to the ants of California [59]. 

All arthropods are stored at the Philpott Laboratory at the University of California, Santa Cruz. We used species richness in this paper to refer to the respective level of identification for each taxon (e.g., to species or morphospecies). We summed the number of individuals captured with any sampling method to determine the total abundance for bees and ladybeetles and the activity density of spiders. For ants, we used occurrence (presence of a species in a trap) instead of number of individuals as our abundance metric [60].

### 2.6. Native Arthropods Characterization

To classify sampled arthropods as native or non-native, we used several resources for each taxon (Appendix A). For bees, we used natural history information, distribution maps, and distribution information from three resources [61,62,63]. For species that were only identified at the genus and morphospecies levels, we examined information and distribution maps for all species within each genus found in California. If one of the resources described a genus as restricted to or only found in California, we classified all morphospecies in that genus as native. If all species within a genus found in California were only distributed in N. America, we considered the species to be likely native. For genera for which certain species are distributed across the Holarctic region, or are not restricted to N. America, we considered the species to have an unknown distribution. Bee species were considered to be non-native only if this was confirmed in [61]. Ladybeetle species were classified as native or non-native following [64] and sources therein, and two online resources [52,65]. We classified spiders using three resources [66,67,68]. For spiders identified at species level, we determined whether they were native or not native based on information in these resources. For spiders that were identified at family or morphospecies level within a family, we examined the ‘Pacific Coast Fauna’ sections of the family chapters in [66]. If all members of a family were described as native, we classified the species as native. If members of the family (and genera within that family) were described as having limited distributions restricted to California or the Western states, we classified the species as likely native. For spider families with large numbers of poorly known genera, or confirmed introduced and native species, we classified these species as unknown. We determined the native status of ants with information from two sources [59,69] and references therein. 

### 2.7. Data Analysis

We used generalized linear models (GLM) or generalized linear mixed models (GLMM) to examine the relationships between the abundance and richness of bees, ladybeetles, ants, and spiders and the local and landscape characteristics of urban gardens. For the two groups sampled in two years (bees and ladybeetles), we used GLMM, and for the groups sampled in only one year (spiders and ants), we used GLM. For each arthropod taxon, we included six dependent variables: (a) the abundance (or number of occurrences) of all species, (b) abundance of native species, (c) abundance of non-native species, (d) species richness of all species, (e) species richness of native species, and (f) species richness of non-native species. For bees and spiders, where many organisms were only identified to the morphospecies level, we included two additional variables: (e) the abundance of native species plus likely native species, and (f) the richness of native species plus likely native species. Thus, we included both a conservative (only species confirmed to be native) and a more inclusive (species confirmed to be native plus species likely to be native) metric for native species abundance and richness in our sites. For our predictor variables, we chose six local scale variables (garden size, number of native plant species, mulch cover within 1 m, number of flowers, number of trees and shrubs) and one landscape variable (percent of natural land cover within 2 km) that represent the variation in garden ground cover and vegetation characteristics and landscape cover and that have been used in other urban garden studies [44]. We only used vegetation data that corresponded to the same year in which arthropods were sampled (i.e., we used 2013 vegetation data as predictors of 2013 arthropod data). We used a variance inflation factor (VIF) cutoff of 3 to select predictor variables that were not highly correlated or collinear with each other. We used natural-log-transformed variables for garden size, number of native plant species, number of flowers, and number of trees and shrubs. Prior to running the models, we used the ‘DHARMa’ package in R to determine the best error distribution for each model based on visual assessment of QQ and standardized residual plots [70]. We conducted all analyses in R statistical software version 4.2.2 [71].

For GLMM, we used the ‘dredge’ function in the ‘MuMIn’ package version 1.42.1 [72] to run all iterations of predictor variables, and ran model selection with the AIC scores to select the best models. If any models were within 2 AIC scores of the best model, we used the ‘model.avg’ function to average the top models. We used a Gaussian distribution for all models. For generalized linear models (GLMs), we tested all combinations of the six explanatory variables with the ‘glmulti’ function [73] and selected the top model based on the AICc values. For all models where the AICc value was within two points of the next best model, we averaged them with the ‘model.avg’ function in the ‘MuMIn’ package [72] and reported conditional averages for significant model factors. We used a Gaussian distribution for all spider abundance variables, and a Poisson distribution for all spider richness variables. For ants, we used a Poisson distribution for the number of ant occurrences, a negative binomial for the number of native ant occurrences, and a Gaussian distribution for both ant richness variables. We visualized all significant predictors of arthropod abundance and richness for both GLM and GLMM models with ‘visreg’ in R [74]. 

All results for the conservative and more inclusive metrics of native bee and spider abundance and species richness were qualitatively similar, and thus we only report the results for the conservative metric here. We also tested all models using observed species richness and estimated species richness with the Chao2 species richness estimator, and the results were qualitatively similar, so we report the output for observed species richness. 

## 3. Results

### 3.1. Summary Data

Between 2013 and 2015, we recorded a total of 199 herbaceous plant species, among which 20 (10.1%) were native species. Native plant richness per year per site in a given year ranged from zero to eight species (or between 0% and 33.3% of species encountered). The percentage of plant cover from native plants ranged from 0% to 49.0%, with just one site having more than 14.2% cover from natives during one year. On average, native plants accounted for 2.53% of all plant cover in garden sites. The most commonly encountered native plant species were *Helianthus annuus* (sunflower, representing 39.0% of all native plant cover), *Eschscholzia californica* (California poppy, 28.6%), *Delphinium* sp. (larkspur, 7.1%), *Achillea millefolium* (yarrow, 5.6%), and *Epilobium* sp. (fireweed, 3.9%) (Appendix A). These five species accounted for nearly 75% of all plant cover from native species.

We collected 70 native species or morphospecies of bees, ladybeetles, spiders, and ants, 51 species that are likely to be native, 17 species that are non-native, and 36 species of unknown origin (Appendix A). We collected 100 bee species and morphospecies from 5753 individuals. Of those, 43 species (and 2580 individuals) are native, 36 species (and 458 individuals) are likely native, and 16 species (and 84 individuals) could not be confirmed as native or not native; all other species and individuals are non-native. We collected 20 ladybeetle species from 1566 individuals. Of those, 17 species (and 1146 individuals) are native, and all other species and individuals are not native. We collected 46 spider morphospecies from 1532 spider individuals. Of those, 6 species (and 1093 individuals) are native species, 15 species (and 143 individuals) are likely native, and 20 species (and 78 species) could not be confirmed to be native or not native. We collected 8 ant species from 726 occurrences; of those, 4 species and 123 occurrences are native, and all others are not native.

### 3.2. Predictors of Arthropod and Native Arthropod Abundance and Richness

We found that bee abundance and richness were influenced by garden size and the amount of natural habitat within 2 km. Total bee abundance and richness were higher in larger gardens (Table 1 and Appendix A), native bee abundance and richness were higher in larger gardens (Figure 2A,B, Table 1 and Appendix A), and the abundance of non-native bee species was higher in larger gardens (Figure 2C) and with more natural habitat in the landscape (Figure 2D, Table 1 and Appendix A), but non-native bee richness did not vary with any local or landscape factors (Table 1 and Appendix A). 

Ladybeetle abundance and species richness responded to changes in the number of trees and shrubs in a garden and natural habitat within 2 km. Total ladybeetle richness was higher in gardens with more trees and shrubs, while total ladybeetle abundance and richness were both lower in gardens surrounded by more natural habitat in the landscape (Table 1 and Appendix A). Native ladybeetle abundance did not differ depending on garden or landscape factors, but native ladybeetle species richness was lower in gardens surrounded by more natural habitat (Figure 2E, Table 1 and Appendix A). The abundance and richness of non-native ladybeetles did not differ with garden or landscape factors (Table 1 and Appendix A).

Spider abundance and richness responded to all local factors, including changes in native plant richness, garden size, floral abundance, tree and shrub abundance, and mulch cover, but no landscape factors. Total spider abundance did not vary with any local or landscape factors, but total spider richness was higher in gardens with more flowers per plot (Table 1 and Appendix A). Native spider richness increased with floral abundance (Figure 2F), but declined with native plant richness (Figure 2G), while native spider abundance did not respond to local or landscape factors (Table 1 and Appendix A). Non-native spider abundance increased with floral abundance (Figure 2H) and non-native spider richness increased with mulch cover (Figure 2I), garden size (Figure 2J), floral abundance (Figure 2K), and tree and shrub abundance (Figure 2L) (Table 1 and Appendix A). 

Ant occurrence and richness were largely unaffected by changes in local and landscape characteristics of gardens. Only the number of native ant occurrences was higher with natural habitat in the landscape (Figure 2M), with no other ant variables responding to either local or landscape predictors (Table 1 and Appendix A). 

## 4. Discussion

Overall, we document that different arthropod taxa (bees, ladybeetles, ants, and ground-foraging spiders) and different groups of arthropods (native vs. non-native) respond to garden and landscape-level natural habitat cover in different ways, but native plant species richness did not have large impacts on most groups of arthropods. Although garden size, floral abundance, tree and shrub abundance, and mulch cover were always positively correlated with arthropod abundance and richness, when significant, native plant richness had only negative impacts (on just one group) and natural habitat had mixed effects on arthropod abundance and richness (e.g., sometimes positive, sometimes negative) depending on the taxon or group. Although we did document significant correlations between the overall abundance and richness of arthropods and local and landscape factors for some arthropod taxa, we focus our discussion here on the effects of native plants, as well as the effects of local and landscape characteristics, on native and non-native arthropods in the gardens. The results of the local and landscape drivers of overall abundance and richness of bees [75], ladybeetles [76], and spiders [77] are discussed in other publications. 

Native plant richness did not impact the abundance or richness of bees, ladybeetles, or ants, and was negatively associated with native spider richness, but did not influence other spider variables. This finding is consistent with many other studies summarized in a recent meta-analysis that failed to document the impacts of native plants on arthropods (17% of studies included), or that found mixed effects of native plants on animal diversity (33% of studies) within urban ecosystems, even if more studies do find positive relationships between native plants and animal diversity (43% of studies) [32]. There are several possible reasons for the lack of effects of native plants on garden arthropods. First, native plant species richness only ranged from zero to eight species, and on average, native plants only accounted for 2.5% of herbaceous plant cover in gardens, with most sites ranging from 0% to 15% of herbaceous plant cover from native plants. Of the limited native plant cover in our sites, more than two-thirds was accounted for by just two species—sunflower (39.0% of native plant cover) and California poppy (28.6%). The California poppy provides pollen resources to bees, but does not produce nectar [78], likely making this a less attractive species for the other arthropods examined in this study. Even though one site had relatively high native herbaceous plant cover (49.0%), that same site had only 4% vegetative cover in the garden, and only six plant species in total, making it far less vegetated than any other site. Thus, the main reason that native plant species may have not been important in this study because they represent a small fraction of the plant species (~10%) and a very small fraction of the total plant cover, consistent with other studies that have assessed native plants within community gardens in California [18]. Second, although native plant richness, biomass, and percent cover are more often used as metrics to assess the importance of native plants for arthropods [32], native floral richness or abundance may be a better predictor of the role native plants play in urban agroecosystems [79]. Unfortunately, we did not count flowers on individual native plants during our study, but rather counted the total number of flowers in the 1 × 1 m study plots, so we cannot assess the role of floral abundance from native plants. Third, we found relatively small proportions of native insects in our study system, with about 40% confirmed native insect species and 29% likely native insect species. In contrast, a study of urban greenspaces in Australia reporting strong positive associations between native plants and insect biodiversity found over 96% native insect species [80]. One important note is that beyond species-level diversity, composition, or the origin of plant species, it is vital to understand the functional diversity and composition of plant traits, such as growth form, floral attraction, and the presence of structures such as trichomes, spines, and nectaries, as such factors can critically impact arthropods such as pollinators and natural enemies [81,82,83], and may be more important than species identity. For bees in particular, our only sampled group of plant mutualists, co-evolutionary processes may shape their interactions with native plants [84]. For instance, some bee species may preferentially collect pollen from co-evolved plants or from phylogenetically related plants in novel environments [85], while other species may heavily rely on non-native plants due to their higher pollen protein resources compared with native congeners [86]. Thus, several detailed mechanisms might contribute to how bees interact with native plant species in urban gardens that deserve further exploration. We did find that native plant abundance negatively affected native spider richness. In contrast, others have found higher native spider richness in crops next to native plantings (shelterbelts) [87] as well as higher overall spider abundance in native vegetation adjacent to crops [88], but these studies focus on patches of native vegetation rather than native plants interspersed in a community garden. Furthermore, these two studies focus on vegetation-foraging spiders, whereas our sample method focused on ground-foraging spiders. In addition, the pattern relating native spider richness to native plant richness in our study seems to be driven by a relatively higher native species richness in one site; removing that site from this particular analysis eliminated the significant relationship. Thus, in general, native plants did not play a big role in supporting overall or native biodiversity in the study sites. 

Other garden features, such as garden size, floral abundance, mulch cover, and the number of trees and shrubs, were important positive predictors of abundance and richness of several arthropod groups. The abundance of native and non-native bees and the richness of native bees all increased with garden size, as did the richness of non-native spider species. Larger gardens likely supply more food or nesting resources for arthropods, and other studies on urban bees document that lower resource availability negatively affects bees [79,89]. Conversely, increases in both floral abundance and the availability of nest sites, both of which may increase in larger gardens, can boost bee abundance [90,91,92]. More specifically, other studies on urban bees have found that larger gardens support more bee individuals [30]. These same resource patterns may be boosting the richness of non-native spider species by providing more prey, nectar or pollen sources, and refuges. Floral abundance boosted native and non-native spider richness as well as non-native spider abundance. Spiders may benefit from nectar and pollen on flowers [93], or potentially may benefit from increased prey abundance of other organisms that use the floral resources. We also found that mulch cover boosted non-native spider abundance. The most abundant non-native spiders were also the second, third, and fourth most abundant spider species overall, with *Trachyzelotes barbatus* (5.3% of all spiders collected), *Urozelotes rusticus* (4.4%), and *Dysdera crocata* (4.3%) accounting for over 98% of all non-native individuals encountered. While information is lacking on the specific influence of mulch on these species, mulch can boost spider abundance generally [94] because it enhances the architectural complexity of agricultural sites [95]. Mulch may also retain soil moisture, benefitting spiders [96], or may lower predation risk for spiders due to the camouflage that mulch provides to ground foragers [97]. Finally, tree and shrub abundance was associated with higher non-native spider species richness. The most common non-native spiders found here are all active hunters that may be using woody vegetation as nesting habitat or foraging substrates. Moreover, other studies have found tree and shrub cover to be a driver of spider richness in urban green spaces [98].

We found that natural habitat had a positive effect on the abundance of both non-native bees and native ants, but a negative effect on native ladybeetle richness. Natural habitats, including woody areas and grasslands, can provide resources to both native and non-native bees, and various studies have documented that honeybees, our most abundant non-native bee, frequently use natural habitats as a primary [99] or seasonal [100] sources of pollen and nectar from floral resources. Thus, a higher abundance of natural habitat may support increased resources and increased populations for non-native bees with larger foraging ranges. We collected four species of native ants—three were found from among fewer than 16 individuals (e.g., *Formica moki*, *Prenolepis imparis*, and *Tapinoma sessile*), and thus are unlikely to be driving the observed patterns for native ant abundance. The only native ant that was common (from 358 individuals) was *Hypoponera opacior*. This species is a small, predaceous ant that nests in soil, under rocks, and in leaf litter, and prefers shady, dryer areas [59,101,102]—all characteristics that describe native habitats such as forests or native grasslands in the study area. Consistent with our study, another study that examined ant communities along an urban to rural gradient near to our study sites found that native ant species were far more common in natural habitat types, or those sites furthest and least disturbed by humans, and that the same non-native ant species that we collected were common in semi-natural, urban, and agricultural habitats [69]. They also found that the abundance of non-native ant species, such as Argentine ant (*Linepithema humile*), was most associated with increases in concrete cover, which tends to be negatively correlated with natural habitat in our study sites. In our study, the Argentine ant accounted for 50.8% of all ant occurrences (and 81.3% of individuals encountered), and they are widely known to displace native organisms [103,104,105] and have strong influences on ant communities in urban and agricultural sites [106]. None of the local garden management or landscape features examined in this study influenced the number of Argentine ant occurrences (Appendix A), but we cannot rule out the possibility that their abundance within the sites influenced the native ant occurrences. Thus, both habitat preferences for natural habitat and negative interactions with Argentine ants may be driving our observed patterns for native ant abundance. The richness of native ladybeetles dropped in sites with more natural habitat, a result in contrast with another study that found that native ladybeetle diversity was positively related to forest cover in the landscape [107]. The most common non-native ladybeetle in our study, *Harmonia axyridis*, represented nearly 76% of all non-native individuals, and this species can have negative impacts on populations of native ladybeetle species (e.g., [108]). Thus, we also considered whether any of our garden or landscape features influenced the abundance of this species. We found that *H. axyridis* abundance declined in gardens with more natural habitat, but was not influenced by any of the garden management features studied (Appendix A, Appendix A). Given that we also found declines in native ladybeetle richness with natural habitat, and that *H. axyridis* only represented 5% of all ladybeetles encountered, it is not likely that interactions with *H. axyridis* are driving the observed patterns. Finally, in our system, the most common native ladybeetle species was *Psyllobora vigintimaculata,* a mycophagous species that feeds on powdery mildew. Powdery mildew is common on many garden crops, and *P. vigintimaculata* densities have been reported to increase with the incidence of powdery mildew in urban greenspaces [109]. A lower incidence of powdery mildew on the trees, scrubs, and grasses found in natural habitats may be an underlying factor driving lower native ladybeetle abundance here.

From a strictly conservation perspective, supporting native biodiversity—of plants and of arthropods—may be an important goal [110,111]. However, some non-native organisms may have conservation value [112], and there may be complex relationships between native and non-native species and the provisioning of ecosystem services [113,114]. Within urban gardens, ecosystem services, such as pollination and pest control, may be provided by both native and non-native species. While native bee abundance and richness may be important for pollination in urban gardens [79,115,116], honeybees (*Apis mellifera*) are still the most abundant bees in our study sites, account for large numbers of floral visits in gardens [117] and likely provide pollination services to crop and non-crop plants therein. Moreover, Argentine ants are by far the most efficient predators of some species of sentinel prey within gardens [118], and non-native ladybeetles can still provide effective pest control in agroecosystems including in gardens [119]. Thus, a focus on boosting native species as well as harnessing the services from non-native species could be beneficial for gardeners. Consistently, larger gardens, floral abundance, and mulch cover benefitted garden arthropods, whereas native plant cover had minimal effects, and natural habitat had mixed effects depending on the arthropod group examined. Thus, for gardeners, focusing on floral abundance may be a more important strategy for promoting biodiversity and services in urban agroecosystems. That said, there is relatively little work examining how the restoration of native plants within urban gardens or a higher fraction of native plant cover within gardens may serve to conserve arthropods, and especially native arthropods. Thus, continuing to study the impact of native plants and wildflowers, as well as the negative impacts of non-native and invasive species on other members of the arthropod community and food web is warranted.

## Figures and Tables

**Figure 1 insects-14-00576-f001:**
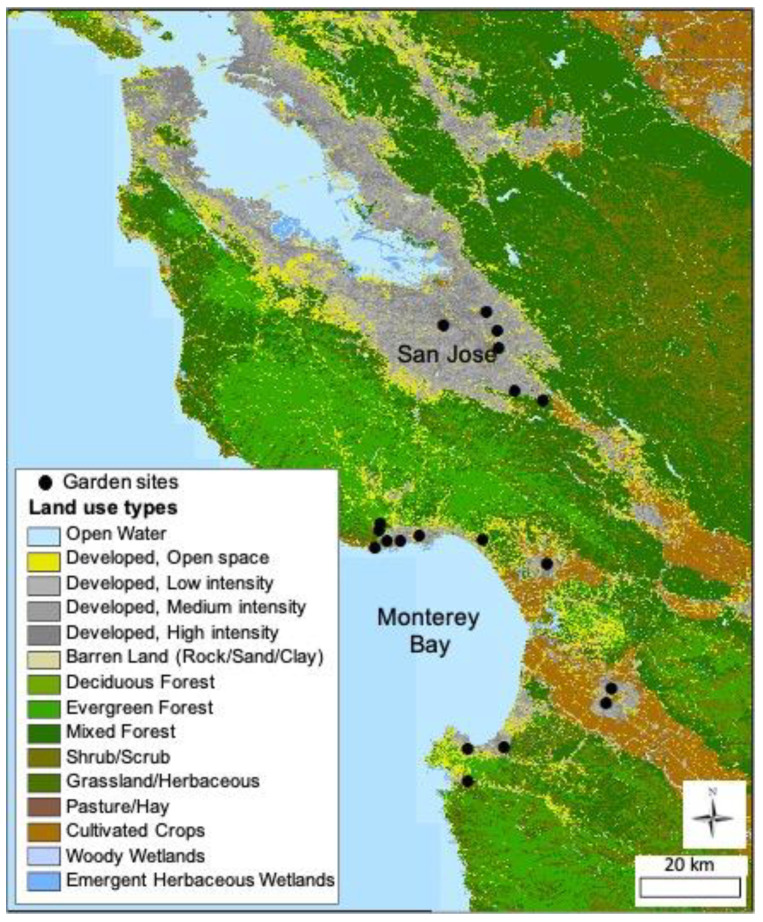
A map of 19 urban garden sites located along the central coast of California.

**Figure 2 insects-14-00576-f002:**
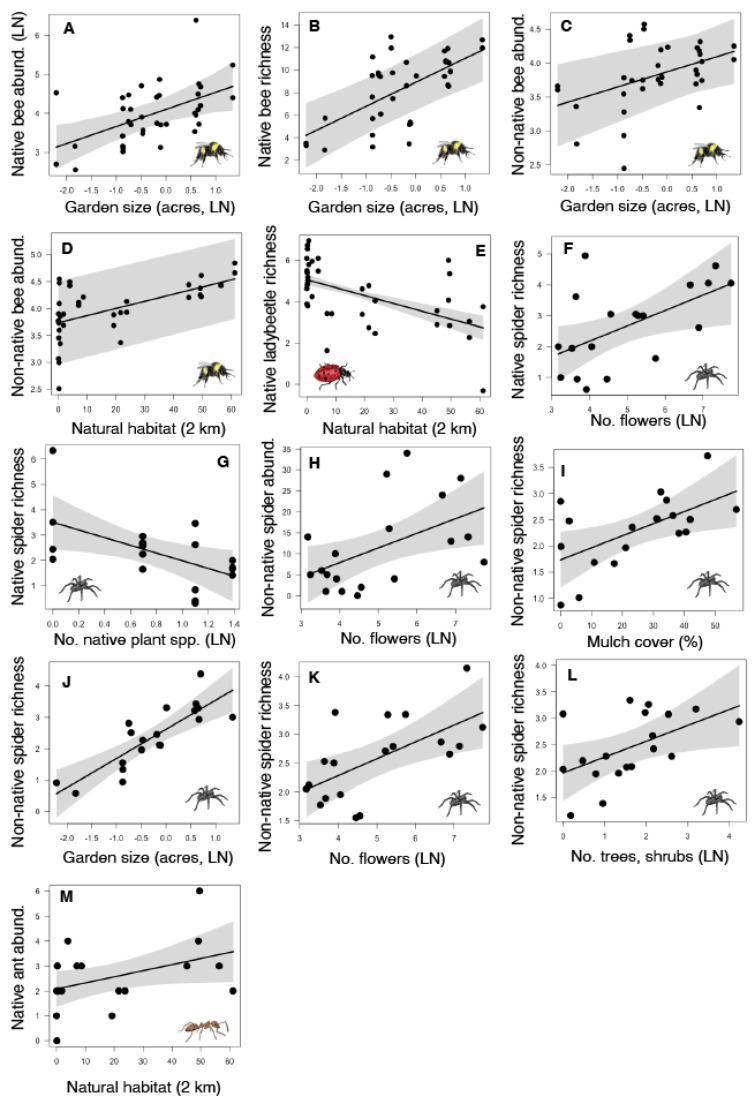
Significant relationships between arthropod abundance (number of individuals of bees, ladybeetles, and spiders, number of occurrences of ants) and richness (number of species) and native plants, other local garden characteristics (e.g., garden size, number of flowers per plot, number of trees and shrubs per plot, average mulch cover), and one landscape factor (natural habitat cover within 2 km) in urban gardens in the California Central Coast region. Figure panels show significant (*p* < 0.05) predictors for bee abundance and richness (**A**–**D**), ladybeetle richness (**E**), spider abundance and richness (**F**–**L**), and ant abundance (**M**). Each black dot represents data from one garden, black lines show the best fit according to statistical models, and grey bands show 95% confidence intervals around the best fit lines.

**Table 1 insects-14-00576-t001:** Summary of correlations between abundance (number of individuals of bees, ladybeetles, and spiders, number of occurrences of ants) and richness (number of species) of all, native, and non-native bees, ladybeetles, spiders, and ants and native plant species richness, other local garden characteristics (e.g., garden size, number of flowers per plot, number of trees and shrubs per plot, average mulch cover), and one landscape factor (natural habitat cover within 2 km) in urban gardens in the California Central Coast region ^1^.

Taxon	Metric	Status	Native Plant Species	Garden Size	No. Flowers	No. Trees, Shrubs	Mulch Cover (1 m)	Natural Habitat (2 km)
Bees 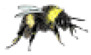	Abundance	All	ns	**+**	ns	ns	ns	ns
Native	ns	**+**	ns	ns	ns	ns
Non-Native	ns	**+**	ns	ns	ns	**+**
Richness	All	ns	**+**	ns	ns	ns	ns
Native	ns	**+**	ns	ns	ns	ns
Non-Native	ns	ns	ns	ns	ns	ns
Ladybeetles 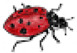	Abundance	All	ns	ns	ns	ns	ns	**−**
Native	ns	ns	ns	ns	ns	ns
Non-Native	ns	ns	ns	ns	ns	ns
Richness	All	ns	ns	ns	**+**	ns	**−**
Native	ns	ns	ns	ns	ns	**−**
Non-Native	ns	ns	ns	ns	ns	ns
Spiders 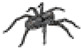	Abundance	All	ns	ns	ns	ns	ns	ns
Native	ns	ns	ns	ns	ns	ns
Non-Native	ns	ns	**+**	ns	ns	ns
Richness	All	ns	ns	**+**	ns	ns	ns
Native	**−**	ns	**+**	ns	ns	ns
Non-Native	ns	**+**	**+**	**+**	**+**	ns
Ants 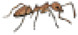	Abundance	All	ns	ns	ns	ns	ns	ns
Native	ns	ns	ns	ns	ns	**+**
Non-Native	ns	ns	ns	ns	ns	ns
Richness	All	ns	ns	ns	ns	ns	ns
Native	ns	ns	ns	ns	ns	ns
Non-Native	ns	ns	ns	ns	ns	ns

^1^ Significant positive correlations are indicated by “**+**”, significant negative correlations are indicated by “**−**”, and non-significant relationships are indicated by “ns”. See Appendix A for the full GLMM model output, statistical values, and effect sizes.

## Data Availability

The summary data used for this article is available at Dryad.org at the following DOI: doi:10.7291/D1HD59.

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
