# Peer review of "Promoting Beneficial Arthropods in Urban Agroecosystems: Focus on Flowers, Maybe Not Native Plants"

_insects, 2023, doi:10.3390/insects14070576_

Round 1
Reviewer 1 Report (New Reviewer)
Promoting beneficial arthropods in urban agroecosystems: focus on flowers, not native plants
In this manuscript, the authors analysed urban community gardens on a landscape scale, considering local parameters such as garden size, flowers, mulch, trees, and shrubs, to assess their impact on native and non-native arthropods. Based on the importance of preventing arthropod decline, understanding how to protect them within cities, specifically in urban gardens, is of great importance. Despite my enthusiasm about this manuscript, I came across a couple of major issues that I would like to see fixed and some minor comments.
Major comments
My main concern about this manuscript is that it primarily focuses on the impact of native plants on arthropods. Nevertheless, it appears that the cover (2.53%) and diversity (2-5 dominant species) of the native vegetation were very low. Based on this, the authors cannot draw significant conclusions.
Furthermore, the authors concentrated on native and non-native arthropods but did not consider different invasive arthropods, which form a non-negligible category. I would like to understand why? For instance, species such as Harmonia axyridis and Linepithema humile could have been examined to determine which local factors may support invasive species. (Invasive species: L. humile has been found to compete with native ants in San Luis Obispo, California, causing significant declines in the populations of 10 native ant species - Kennedy, T. A. (1998). Patterns of an invasion by Argentine ants (Linepithema humile) in a riparian corridor and its effects on ant diversity. American Midland Naturalist, 343-350.) – can be interesting.
Also, I missed the exact species data.
Why did the authors not study the type of garden management of the community gardens too?
Most parts of the manuscript read quite well.
Minor comments
Title: I suggest change the title, depend on your main results and message.
Introduction: I recommend to add a paragraph about community gardens and the importance of agroecosystems.
Table 1. Change the table and numerically show the statistical values and effect size.
L403 Do you study the grass too, or just flowers? Here needs to be clarified.
L87 I suggest to add information about agricultural practices and management types.
Conclusion: It would be beneficial to mention that there is a lack of research on the restoration of wildflowers and native plant species in gardens. Investigating the impact of this restoration is crucial because it is premature to draw a definitive conclusion that non-natives support arthropods. Additionally, non-native arthropods can pose significant problems, as their effects on the native arthropod community and food web need to be thoroughly understood over the long term.
Author Response
Comments and Suggestions for Authors
Promoting beneficial arthropods in urban agroecosystems: focus on flowers, not native plants
In this manuscript, the authors analysed urban community gardens on a landscape scale, considering local parameters such as garden size, flowers, mulch, trees, and shrubs, to assess their impact on native and non-native arthropods. Based on the importance of preventing arthropod decline, understanding how to protect them within cities, specifically in urban gardens, is of great importance. Despite my enthusiasm about this manuscript, I came across a couple of major issues that I would like to see fixed and some minor comments.
Major comments
My main concern about this manuscript is that it primarily focuses on the impact of native plants on arthropods. Nevertheless, it appears that the cover (2.53%) and diversity (2-5 dominant species) of the native vegetation were very low. Based on this, the authors cannot draw significant conclusions.
----- Of course, at the outset of the study we did not know that the plant cover from native plant species would be so low. The species richness was 10% (which is not entirely negligible). We already in our previous submission included a section in the discussion to address that the low native plant cover is likely the main reason that we did not find strong significant effects of native plants. We have now added phrases to the simple summary and abstract to make this same point. Because understanding the role of native plants is the whole premise of this study, we need to make some relevant conclusions to our research questions, and we already do so with the caveats that we mention.
Furthermore, the authors concentrated on native and non-native arthropods but did not consider different invasive arthropods, which form a non-negligible category. I would like to understand why? For instance, species such as Harmonia axyridis and Linepithema humile could have been examined to determine which local factors may support invasive species. (Invasive species: L. humile has been found to compete with native ants in San Luis Obispo, California, causing significant declines in the populations of 10 native ant species - Kennedy, T. A. (1998). Patterns of an invasion by Argentine ants (Linepithema humile) in a riparian corridor and its effects on ant diversity. American Midland Naturalist, 343-350.) – can be interesting.
---- This is an interesting suggestion. We did, after reading this comment, perform an analysis on how the same studied factors used in the rest of the study (e.g., garden size, number of native plant species, natural habitat in the landscape) influenced the number of Linepithema humile occurrences and Harmonia axyridis individuals.
------ For L. humile, no factors predicted the number of occurrences (See revised Table S3). We have added a sentence to the discussion to show that we performed this analysis and to re-emphasize the point that we made in our earlier draft that the abundance of this ant species has negative impacts on native ants. We have added the reference suggested by the reviewer. Specifically, we added, “None of the local garden management or landscape features examined in this study influenced the number of Argentine ant occurrences (Table S3), but we cannot rule out the possibility that their abundance within the sites influenced the native and occurrences.”
----- We also ran the analysis for Harmonia axyridis. We note that this species represented only about 5% of the ladybug individuals captured, which we know is surprising given how common they are in some disturbed habitats across N. America. However, the species did represent nearly 76% of the non-native ladybugs. So, we suspect that some of the patterns observed for non-native ladybug abundance might be strongly driven by this species. Moreover, H. axyridis could of course be influencing the native ladybug species. We actually found a negative correlation between H. axyridis and natural habitat in the landscape (see revised Table S3, and new Figure S2), but none of the local garden management features that we measured influence the abundance of this species. We have added this text to the discussion section: “The most common non-native ladybeetle in our study, Harmonia axyridis represented nearly 76% of all non-native individuals, and this species can have negative impacts on populations of native ladybeetle species (e.g., [97]). Thus, we also considered whether any of our garden or landscape features influenced the abundance of this species. We found that H. axyridis abundance declined in gardens with more natural habitat, but was not influenced by any garden management feature studied (Figure S2, Table S3). Given that we also found declines in native ladybeetle richness with natural habitat, and that H. axyridis only represented 5% of all ladybeetles encountered, it is this is not likely that interactions with H. axyridis are driving the observed patterns.
--- We add this information to the discussion rather than in the main methods and results because a species by species analysis was not the intent of this study. We do think it is interesting, as the reviewer points out, to consider impacts on prominent non-native (or in this case invasive) species, but since this was not the specific goal of the study (to examine impacts of invasives on the native species), we feel we are limited in what we can conclude.
Also, I missed the exact species data.
----- The species data is presented in Table S1. We included this table in each version of the manuscript we have submitted. It lists every single species and morphospecies and their abundances in the project. If there is other exact species data that the reviewer is looking for, we would be happy to provide this information.
Why did the authors not study the type of garden management of the community gardens too?
---- Indeed, we have included extensive analysis on the type of garden management in the gardens in this study, and report on these factors throughout the manuscript. We have collected data on vegetation and ground cover characteristics including canopy cover, number of trees and shrub individuals and species, number of tree and shrub individuals in flower, the number of herbaceous plant species, the number of herbaceous plant flowers, herbaceous plant height, as well as ground cover provided by bare ground, grass, herbaceous plants, rocks, wood, leaf litter, mulch and concrete as outlined in section 2.2. These changes in herbaceous and woody vegetation and ground cover are among the most important characteristics of garden management. We also report on garden size (which in our sties is consistently correlated with garden age). We did not use every single one of these metrics in our analysis, as many are co-linear, but we do include representative variables in the data analysis. There are certainly many other characteristics of garden management (plant composition, presence of trellises, soil amendments, demographics of the gardeners, etc.) that we did not measure (and cannot measure now many years after the study), but in our assessment, we did measure a very large number of variables associated with garden management.
Most parts of the manuscript read quite well.
---- We thank the reviewer for this positive comment.
Minor comments
Title: I suggest change the title, depend on your main results and message.
-----We have changed the title to: Promoting beneficial arthropods in urban agroecosystems: focus on flowers, maybe not native plants
Introduction: I recommend to add a paragraph about community gardens and the importance of agroecosystems.
---- We have added a paragraph to the introduction.
Table 1. Change the table and numerically show the statistical values and effect size.
---- All of the statistical output is available in Table S3 (including the full GLMM output, estimates, Z or t-scores, and p-values). Our intent in providing the Table 1 in the main text and Table S3 in the supplement was to keep the main manuscript more digestible, but still have the detailed output for those who are interested in the values. We have added to the Table 1 footnote, “See Table S3 for full GLMM model output, statistical values, and effect sizes. We know that space in journals is highly limited, thus we will be happy to add this information to the main text, if the journal editors encourage us to do so.
L403 Do you study the grass too, or just flowers? Here needs to be clarified.
----Thanks for asking. We think here the reviewer is referring to grass and flowering plants? We did estimate the percent cover from grasses (which was ~4% of herbaceous plant cover), but did not identify grasses to species, and we did not include cover from grass in the denominator of “herbaceous plant cover”. We have added clarifications to sections 2.2 and 2.3 to clarify this.
L87 I suggest to add information about agricultural practices and management types.
----- We apologize, but we cannot understand which text this is referring to or where the reviewer would like us to add this text. Line 87 (in the previous manuscript version) is a section that is describing the impacts of native plants on organisms. It is not a section talking about urban agroecosystems or garden management. In our paragraph on urban agroecosystems, we do refer to review articles that summarize a lot of literature on the impacts of garden management on biodiversity.
Conclusion: It would be beneficial to mention that there is a lack of research on the restoration of wildflowers and native plant species in gardens. Investigating the impact of this restoration is crucial because it is premature to draw a definitive conclusion that non-natives support arthropods. Additionally, non-native arthropods can pose significant problems, as their effects on the native arthropod community and food web need to be thoroughly understood over the long term.
------ We have added this to the end of the manuscript, “That said, there is relatively little work examining how the restoration of native plants within urban gardens, or a higher fraction of native plant cover within gardens might serve to conserve arthropods, and especially native arthropods. Thus, continuing to study the impact of native plants and wildflowers, as well as the negative impacts of non-native and invasive species on other members of the arthropod community and food web is warranted. “
Reviewer 2 Report (Previous Reviewer 1)
This is a significant improvement and the authors have addressed all the concerns I expressed in the first review. Given that up to a third of the bee fauna in many cities are specialist, the need to incorporate native plantings should be considered.
Author Response
Comments and Suggestions for Authors
This is a significant improvement and the authors have addressed all the concerns I expressed in the first review. Given that up to a third of the bee fauna in many cities are specialist, the need to incorporate native plantings should be considered.
--- We thank the reviewer for this comment. We have already added a few sentences in the conclusions to state that continued work in this area (and examining restoration of native plants) is important.
Reviewer 3 Report (Previous Reviewer 2)
All the issues identified in my previous review have been answered in a clarifying and satisfying way, so I consider the article suitable for publication. The only thing that remains to be clarified throughout the text and figures is that the spiders studied in the article are ground-foraging (or soil) spiders (L13, L30 and throughout the text).
Author Response
Comments and Suggestions for Authors
All the issues identified in my previous review have been answered in a clarifying and satisfying way, so I consider the article suitable for publication. The only thing that remains to be clarified throughout the text and figures is that the spiders studied in the article are ground-foraging (or soil) spiders (L13, L30 and throughout the text).
--- Thanks for the positive assessment. We have added “ground-foraging” spiders in several instances, and in the methods say “ground-foraging spiders (hereafter spiders)” so as to save space in the journal.
Round 2
Reviewer 1 Report (New Reviewer)
Second round of “Promoting beneficial arthropods in urban agroecosystems: focus on flowers, not native plants” by Stacy M. Philpott et al., manuscript number insects-2433632.
I am delighted to read the enhanced version of this manuscript. The authors have addressed the questions and incorporated the proposed changes into the manuscript. Below, I have a few minor comments:
I suggest replacing 'urban agroecosystems' with the more specific term 'community gardens.' In the abstract part.
I am still interested in learning about the management practices employed in the community gardens, such as pesticide usage, frequency of mowing, and I recommend adding information about the material part.
Finally, it would be a great idea to consider changing the title to better reflect the main message of your study.
Author Response
Second round of “Promoting beneficial arthropods in urban agroecosystems: focus on flowers, not native plants” by Stacy M. Philpott et al., manuscript number insects-2433632.
I am delighted to read the enhanced version of this manuscript. The authors have addressed the questions and incorporated the proposed changes into the manuscript.
----- We thank the reviewer for their positive assessment
Below, I have a few minor comments:
I suggest replacing 'urban agroecosystems' with the more specific term 'community gardens.' In the abstract part.
----- We have added “urban community garden” in the abstract the first time that we mention urban agroecosystem
I am still interested in learning about the management practices employed in the community gardens, such as pesticide usage, frequency of mowing, and I recommend adding information about the material part.
---- We have added this information into section 2.1. The specific text is, “The gardens are all managed using organic practices, and none allow chemical pesticides or herbicides to be used. None of the gardens contain large patches of grass, and none experience regular mowing.”
Finally, it would be a great idea to consider changing the title to better reflect the main message of your study.
---- We have changed the title in the last round of revisions to include “maybe native plants”. Since the focus of studying native plants was the main point of our study, we find it is important to put native plants in the title. Because we didn’t find strong effects, we are not sure what else to put as a title. We are open to suggestions.
This manuscript is a resubmission of an earlier submission. The following is a list of the peer review reports and author responses from that submission.
Round 1
Reviewer 1 Report
Overall, the manuscript presents relevant data and results for the field of urban biodiversity in general and urban ag in specific. The authors propose a more comprehensive approach to the role of insect-plant interactions in urban ag settings than many other authors have presented recently. That is indeed needed in the field. Alternatively, the results and discussion sections focused mostly on individual groups instead of a more community based perspective.
An important question that emerges from the work is how functionally redundant or complimentary is the non-native vegetation to the native? Also, any native cultivars are bred for traits that are attractive to people and not pollinators. Were those present, and if so, were they counted as natives?
L.49 For a more recent, and broader, study on the effects of urbanization on biodiversity I suggest that the authors see:
Simkin, R.D., Seto, K.C., McDonald, R.I. and Jetz, W., 2022. Biodiversity impacts and conservation implications of urban land expansion projected to 2050. Proceedings of the National Academy of Sciences, 119(12), p.e2117297119.
L.62-63 The authors should also consider the role of coevolution as a major driver of pollinator-plant diversity. This is highly relevant given that a non-native might be functionally redundant with a native.
Cardinal, S. and Danforth, B.N., 2013. Bees diversified in the age of eudicots. Proceedings of the Royal Society B: Biological Sciences, 280(1755), p.20122686.
L.70 There are researchers that have found support for the alternative, for example:
Prendergast, K.S., Tomlinson, S., Dixon, K.W., Bateman, P.W. and Menz, M.H., 2022. Urban native vegetation remnants support more diverse native bee communities than residential gardens in Australia's southwest biodiversity hotspot. Biological Conservation, 265, p.109408.
Reviewer 2 Report
Review of the manuscript entitled: Promoting beneficial arthropods in urban agroecosystems: focus on flowers, not native plants.
The manuscript studies the relationship of abundance and species richness of four groups of beneficial arthropods sampled in urban agroecosystems (a topic on which much has been published). It incorporates as a novelty the classification of plants and arthropods into native/non-native, with some subcategories due to difficulties in identification. Abundance and richness of bees, ladybirds, spiders and ants are related to local (plant species richness, size of garden, number of flowers, number of trees or shrubs, mulch cover) and landscape (natural habitats within 2 km) characteristics.
The conclusion drawn from the work is quite important for garden management, even conservation in general and goes against many of the previous publications and assumptions: native plants do not offer better support for natural enemies. Moreover, the quantity and quality of flowering is more important than whether the plants are native.
In my view, the results leading to this conclusion have been obtained with some methods that are not sufficiently clear in the manuscript and which I will detail below.
My main concern is that the spiders, providing the most of the significant relationships (8 out 19, Table 1) with the local and landscape features studied, were sampled in a year (2014) in which, according to the text, garden vegetation was not sampled at the same time (L. 129-133). What values of local and landscape characteristics did the authors use in the spider abundance/richness analysis? Can you be sure that there are no significant differences in these characteristics amongst years? You didn’t show any comparison between the two years the vegetation was sampled: 2013 and 2015.
On the other hand, I think the effect of native/non-native species is not comparable in this study due to the low proportion in terms of abundance, richness and cover of native plants in most of the gardens studied. Even the authors use this argument in discussion (L-379-382) trying to explain the lack of consistent results in the expected way. Furthermore, the predominant native plants found are herbaceous (sunflowers and California popys), whereas I assume that the non-native ones belong to all types of plant architectures. A clarification in this respect would also be useful and moreover you should compare only plants with the same shape, or all architectures should be equally represented in native/non-native plants. The authors also admit in the discussion that they did not count flowers on native plants during the study (L. 386-387), however in Table 1 there is a column for number of flowers, so I interpret that at least this characteristic refers exclusively to non-native plants, as well as the trait number of trees and shrubs.
Another of my questions is about the sampling separation of the different orders of arthropods. I suppose when you are hand netting the plants in 2013 and 2015 sampling bees, a number of ladybirds, spiders and even ants would be also collected. Why this individuals are not used to complete the results? However, in the case of ladybirds, other methods are used, such as direct sampling, when many of the small ones (e.g. Scymnus sp.) are difficult to see with the naked eye, perhaps that is why sampling is supplemented with sticky traps? Why is entomological netting not used for sampling all the orders completing the survey with the other methods?
A drawing or explanation of how pitfall traps are covered with a plate should be provided, I understand that there will be an opening between the plate and the trap to act as a falling trap. Could there be an over-representation in the sampling of ground spiders over those more closely associated with vegetation? Perhaps this fact could influence the unexpected lack of relationship between the abundance of spiders and vegetation traits.
Other comments:
L.151-L161. Please justify the different classification of lawn grass as an open hatitat, grassland as natural habitat and pastures as agricultural habitat. Are there some variables related with animal or plant diversity/richness you consider to differentiate the three types of habitat?
L. 187-190. Why a different number of samplings each year? Meteorological problems?
L. 195. Please use the International System of Units.
L. 334. I am not able to identify the content of this sentence in Table 1.
L. 337. Identify what variables do you consider as local and landscape.
L. 344. Ant abundance or ant occurrence?
L.361-364. I don’t understand well this sentence. Please clarify.
L. 368-369. You use the same reference Berthon et al. 2021 to document the lack of impact of native plants on arthropods in urban ecosystems, the opposite in the Introduction (L. 67 and 77-80), please clarify. I know this is a meta-analysis, but most of the studies included supported the positive effect of native plants on arthropod abundance and richness.
L. 452-455. The study of Gardiner et al.2021 is not consistent with your results of dropping ladybeetle abundance in sites with more natural habitat, as you wrote. In fact they say: “Native coccinellid species richness and native aphidophagous abundance in gardens were positively associated with forest habitat at a landscape scale of 2 km”.
Table 1. I suggest to change nº individuals by abundance and nº species by richness as you use in the text (L. 334). Please homogenize along the text.
Figure 2. The name of the Y axis is confusing, please use abundance and richness terms. I would propose reorder the graphs putting together those pertaining to the same order, and separating with one space (if possible) to the next group.